# Development of Light-Polymerized Dental Composite Resin Reinforced with Electrospun Polyamide Layers

**DOI:** 10.3390/polym15122598

**Published:** 2023-06-07

**Authors:** Aleksandra Maletin, Ivan Ristić, Aleksandra Nešić, Milica Jeremić Knežević, Daniela Đurović Koprivica, Suzana Cakić, Dušica Ilić, Bojana Milekić, Tatjana Puškar, Branka Pilić

**Affiliations:** 1Faculty of Medicine, University of Novi Sad, 21000 Novi Sad, Serbia; milica.jeremic-knezevic@mf.uns.ac.rs (M.J.K.); daniela.djurovic-koprivica@mf.uns.ac.rs (D.Đ.K.); bojana.milekic@mf.uns.ac.rs (B.M.); tatjana.puskar@mf.uns.ac.rs (T.P.); 2Faculty of Technology, University of Novi Sad, 21000 Novi Sad, Serbia; ivan.ristic@uns.ac.rs (I.R.); alexm@uns.ac.rs (A.N.); brapi@uns.ac.rs (B.P.); 3Faculty of Technology, University of Niš, 16000 Leskovac, Serbia; cakics@tf.ni.ac.rs; 4Faculty of Electronic Engineering, University of Niš, 18000 Niš, Serbia; dusica.ilic@elfak.ni.ac.rs

**Keywords:** dental resin composite, electrospinning, polyamide, nanofibers, mechanical properties

## Abstract

As the mechanical properties of resin-based dental composite materials are highly relevant in clinical practice, diverse strategies for their potential enhancement have been proposed in the extant literature, aiming to facilitate their reliable use in dental medicine. In this context, the focus is primarily given to the mechanical properties with the greatest influence on clinical success, i.e., the longevity of the filling in the patient’s mouth and its ability to withstand very strong masticatory forces. Guided by these objectives, the goal of the present study was to ascertain whether the reinforcement of dental composite resins with electrospun polyamide (PA) nanofibers would improve the mechanical strength of dental restoration materials. For this purpose, light-cure dental composite resins were interspersed with one and two layers comprising PA nanofibers in order to investigate the influence of such reinforcement on the mechanical properties of the resulting hybrid resins. One set of the obtained samples was investigated as prepared, while another set was immersed in artificial saliva for 14 days and was subsequently subjected to the same set of analyses, namely Fourier-transform infrared spectroscopy (FTIR), scanning electron microscopy (SEM), and differential scanning calorimetry (DSC). Findings yielded by the FTIR analysis confirmed the structure of the produced dental composite resin material. They also provided evidence that, while the presence of PA nanofibers did not influence the curing process, it strengthened the dental composite resin. Moreover, flexural strength measurements revealed that the inclusion of a 16 μm-thick PA nanolayer enabled the dental composite resin to withstand a load of 3.2 MPa. These findings were supported by the SEM results, which further indicated that immersing the resin in saline solution resulted in a more compact composite material structure. Finally, DSC results indicated that as-prepared as well as saline-treated reinforced samples had a lower glass transition temperature (Tg) compared to pure resin. Specifically, while pure resin had a Tg of 61.6 °C, each additional PA nanolayer decreased the Tg by about 2 °C, while the further reduction was obtained when samples were immersed in saline for 14 days. These results show that electrospinning is a facile method for producing different nanofibers that can be incorporated into resin-based dental composite materials to modify their mechanical properties. Moreover, while their inclusion strengthens the resin-based dental composite materials, it does not affect the course and outcome of the polymerization reaction, which is an important factor for their use in clinical practice.

## 1. Introduction

In modern dental practice, non-invasive caries treatment involves its removal, careful preparation of the remaining cavities, and reconstruction of defects in the tooth structure using suitable restorative material [1]. Typically, this implies a material based on composite resins due to the superior aesthetics, good physical and mechanical properties, and antibacterial and non-toxic effects [2]. Dental composite materials consist of three or more chemically distinct substances that are mutually insoluble and that when combined, yield superior properties to those they would provide individually. These structural components include organic matrix, inorganic filler particles, and bonding agents that facilitate the connection between the organic and inorganic parts, along with other inorganic components that enhance the basic material attributes [3,4].

The composition and properties of dental composite materials have undergone considerable changes since they were first applied in restorative dentistry in the mid-20th century. Owing to the continued research aimed at advancing their characteristics and functionality, not only appropriate aesthetic outcomes but also satisfactory biological, physical, and chemical characteristics have been achieved. Nonetheless, further work is still needed to overcome certain deficiencies related to the mechanical and physical properties of composite materials, such as polymerization shrinkage and insufficient wear resistance, as these issues compromise the longevity and quality of composite restorations in clinical practice [5,6].

Findings yielded by extant dental materials research show that the performance of dental composites can be improved through nanotechnology, which is why most modern dental composite materials contain modified matrices filled with nanofibers [7]. As the mechanical properties of dental composite materials are primarily determined by the content and characteristics of the inorganic component, it is typically modified by incorporating nanofibers with suitable attributes [3]. In this context, nanofiber quantity also plays an important role, as fillers containing nanofibers in small amounts (1−2% of the total mass) exhibit superior mechanical properties, whereas increasing the nanofiber fraction to 4−8% reduces flexural strength, modulus of elasticity, and fracture resistance. Therefore, the desired reinforcement can be attained by including a smaller amount of nanofibers in the filler structure of dental composite materials, while also achieving a more even material distribution and greater resistance to large occlusal forces [8].

In this line of research, findings reported by Ming Tianas well as Hao Fong et al. are particularly relevant, as these authors compared the mechanical properties of composite resins reinforced with one or two layers of 30 μm-thick nylon-6 nanofibers with those of pure resin. According to these authors, a single layer of nylon-6 nanofibers increases flexural strength by 31%, and this value increases to 36% when two layers are adopted. Other authors have similarly noted that reinforcement with electrospun fibers leads to a significant improvement in the mechanical properties of dental composites (such as up to a 3-fold increase in tensile strength) and pointed to the great potential of nanotechnology in this domain [9,10,11].

In extant studies, researchers have also experimented with different particles, aiming to identify those that would strengthen dental composite materials most effectively by modifying their mechanical properties [12]. For example, silicon nanoparticles in different weight ratios [2] or particles of aluminum trioxide [13], alumina-polycarbonate [14], and amorphous calcium phosphate [15] have been previously incorporated into the basic bisphenol A-glycidyl methacrylate (Bis-GMA) organic resin matrix. Based on their analyses, Kumarand co-workers concluded that adding gypsum particles to dental composites exerts a positive influence on its physicomechanical properties and optical characteristics, as well as on the degree of monomer-to-polymer conversion [16].

When dental composite materials were first developed, they were solely utilized for incisor and canine restoration after caries removal. However, owing to technological advances, their use has gradually expanded, whereby they are now routinely applied to posterior teeth (premolars and molars) as well. While this practice results in more aesthetically pleasing results and avoids the use of amalgam fillings that were used in the past, posing considerable health risks to both the patient and the clinician, composite fillings suffer from poor durability. As a result, it is not uncommon for secondary caries, marginal discoloration of the fillings as well as the emergence and spread of cracks in the filling to occur [17,18]. In order to prevent the development of secondary caries without affecting the mechanical properties and biocompatibility of the material, antimicrobial particles such as chitosan and chitosan loaded with dibasic calcium phosphate anhydrous are now typically added to dental composites [19].

With respect to the prevention of cracks, which arise due to the large occlusal forces that are inevitably generated during the normal functioning of the orofacial system, considerable research has been conducted on improving the resistance of composite fillings by adding nanoparticles, nanofibers, and nanotubes to dental composites [20,21]. Available evidence indicates that, as the so-called liquid composites have 20−25% less filler than conventional composites, they exhibit inferior mechanical properties. To overcome this shortcoming, several authors have explored the viability of adding titanium dioxide (TiO_2_) nanotubes to the material structure without compromising its fluidity [22]. These and other fiber-reinforced composites (FRCs) have been developed, with glass, carbon particles, or polyamide (PA) serving as the filler [23,24]. Extensive research has also been conducted on improving the properties of dental materials, including composites. In studies focusing on polymer material technology, electrospinning has been proven the most successful in attaining the desired outcomes [24,25,26].

Electrospinning is a powerful, yet fairly simple and versatile technology that relies on electric forces for producing electrostatic charge among particles, allowing them to accrete in the form of ultrafine fibers [27]. When this method is applied, the electric force acts on the solution containing polymers or other liquid materials to produce a finely dispersed jet, thus leading to the formation of microfibers and nanofibers [28,29]. The resulting ultrafine fibers of various sizes, shapes, and structures are added to a variety of materials in order to improve their performance and expand product capabilities within a number of industrial sectors, including many biomedical fields [29,30].

Thus far, a wide range of materials has been electrospun into nanofibers, thus obtaining the desired morphology, diameter, and surface topology [31,32,33]. This flexibility and versatility are of key importance, as it allows for the properties and orientation of nanofibers to be modified to fit specific requirements [28,33,34]. To achieve this objective, understanding the properties of a single nanofiber is essential because its mechanical performance will determine the mechanical characteristics of nanofiber-incorporated materials [8,26].

Guided by these promising findings, as a part of the present study, the possibility of improving the mechanical properties of resin-based dental composite material by reinforcing it with PA nanofibers, obtained via the electrospinning method, was investigated. For this purpose, polyamide fibers of the desired morphology were first spun in the form of thin films (with different thicknesses) which were subsequently used as reinforcement. This research design was guided by the hypothesis that the addition of PA nanofibers into dental composite materials affects their mechanical properties, as confirmed by mechanical assessments involving the three-point bending test, scanning electron microscopy analysis (SEM), and differential scanning calorimetry analysis (DSC).

## 2. Materials and Methods

Dental composite resin-based restorative material DiaFil was purchased from DiaDent Europe B.V. (Almere, The Netherlands). According to the manufacturer’s specification, it contains the following polymer resins in its composition: urethane dimethacrylate (UDMA, 8−12%), bisphenol A glycidyl methacrylate (Bis-GMA, 6−10%) and triethylene glycol dimethacrylate (TEGDMA, 1−4%), as shown in Table 1. In addition to DiaFil, Polyamide-6 (Akulon-6, DSM, Geleen, The Netherlands) and formic acid (98/100%, Fisher Chemicals, Leicestershire, UK) were procured and were used as received.

### 2.1. Sample Preparation

The 20 wt% of polyamide solution in formic acid required for electrospinning was prepared 24 h in advance by mixing at room temperature using a magnetic stirrer. Electrospun nanofibers were obtained using Fluidnatek LE-10 (Quantum Design GmbH, Darmstadt, Germany) electrospinning apparatus equipped with two syringe pumps (with 10,000 μLh^−1^ maximum feed rate) and high voltage supply (with a maximum voltage of 30 kV). As PA nanolayers of two thicknesses were required for the experiments, 8 μm and 16 μm thick nonwovens were prepared on the plate collector at 200 μLh^−1^ feed rate, 23 kV voltage, and 15 cm tip-to-collector distance. After electrospinning, these nonwovens were cut and shaped for further manipulation. The resulting PA films were denoted as PA8 and PA16, reflecting their 8 μm and 16 μm thickness, respectively.

Three types of resin-based samples that are suitable for mechanical testing were prepared using a metallic mold with 20 mm × 2 mm × 2 mm inner dimensions specifically designed by the manuscript authors for the purposes of this research. The obtained dimensions were carefully checked with Vernier calipers, thus ensuring that all dental composite samples had 20 mm length, 2 mm width, and 2 mm height (within a narrow tolerance range). Thus, all samples that did not meet these stringent criteria were discarded, as it is well known that any deviations could influence the mechanical properties of tested material. The sample preparation process commenced with the casting of pure resin samples, which served as controls. Next, the samples with one and two PA nanolayers were produced, aiming to attain the resin–PA layerstructure. Specifically, five samples of dental composite material with one polyamide fiberlayer of 8 µm thickness (denoted as res-PA8-res), five samples of dental composite material with one polyamide fiberlayer of 16 µm thickness (labeled as res-PA16-res), and five samples of dental composite material with two 8 µm-thick layers of polyamide fibers (denoted as res-PA8-res-PA8-res) were produced, as shown in Table 2.

All samples were cured by applying the SmartLite PS (DentsplayDeTrey GmbH, Konstanz, Germany) UV light lamp of 950 mW/cm^2^ power for 20 s, according to the manufacturer’s instructions. As the manufacturer does not offer a choice of different polymerization modes, all parameters—including the distance between the light source and the sample surface—were always the same.

Once all samples had been prepared and their dimensions had been carefully checked, they were divided into two groups. The first (henceforth denoted as “as-prepared”) group was subjected to analyses 24 h post-curing, while the second (henceforth denoted as “saline-treated”) group was stored for 14 days in hermetically sealed test tubes (Laboratory d.o.o, Novi Sad, Serbia) submerged in a water bath (Memmert GmbH, Schwabach, Germany) that was maintained at a constant temperature of 37 °C. To emulate the conditions prevailing in the patient’s oral cavity, the test tubes contained a physiological solution (NatriiChloridiInfundibile 0.9% NaCl, HemofarmAD, Vršac, Serbia).

### 2.2. FTIR Analysis

All samples were subjected to the FTIR analyses, performed using the IRAffinityATR instrument (Shimadzu, Kyoto, Japan) focusing on the 400–4000 cm^−1^ band range, with 16 scans and 4 cm^−1^ resolution.

### 2.3. Morphological Analyses

For determining the sample morphology, an SEM microscope (JEOLJSM 6460 LV with EDS Oxford INCA, JEOL USA, Inc., Peabody, MA, USA was employed to examine the sample cross-section after fracture during the mechanical tests. Prior to exposing them to the electron beam at 20 kV voltage, samples were secured to the specimen holders with double-sided adhesive tape and were covered with a layer of gold under vacuum.

### 2.4. Mechanical Testing

For assessing the mechanical properties (flexural strength and needle penetration depth), all generated samples were subjected to the three-point bending test performed using the EZ-LX Test (Shimadzu, Kyoto, Japan) tensile testing apparatus. This test is conducted by placing the sample on the holder designed for this purpose, with two supports positioned at a certain distance from each other, while a needle located on the opposite side exerts the pressure on the sample until penetration depth equals 50% of the specified distance has been achieved. In the present study, the speed of the needle penetration was set to 1 mm/min, and the penetration continued until sample fracture occurred.

Based on the recorded measurements, flexural strength (σ, MPa) was calculated based on the following expression:$$\sigma =\frac{3\times L}{2\times b\times h^{2}}\cdot F$$
where

*F*—fracture force [N]

*h*—sample height [mm]

*b*—sample thickness [mm]

*L*—distance between supports [mm]

All measurements were performed in triplicate for each sample and the average values were used in analyses.

### 2.5. Thermal Properties

For determining the thermal properties of the studied samples, differential scanning calorimeter TA Instruments Q20 (Waters, TA Instruments, New Castle, DE, USA) was employed, whereby one heating cycle from 20 to 200 °C at the 10 °C min^−1^ heating rate was adopted. The apparatus was calibrated using gallium and indium as the standards, and nitrogen (at 50 cm^3^/min flow rate) served as the purge gas. The glass transition temperatures were determined using the half Δc_p_ on the calorigrams from the second heating runs.

## 3. Results and Discussion

Electrospun nanofibers have been widely used in dentistry due to their excellent properties, such as large surface area and high porosity. Presently, their application is largely restricted to periodontal regeneration, whereby polycaprolactone (PCL) is usually employed for this purpose. However, owing to the global upsurge in research on electrospun dental materials, bone regeneration, tissue regeneration, and cell differentiation and proliferation, it is expected that the scope of their clinical use will expand considerably [35] Nonetheless, there is a paucity of studies as a part of which the impact of the incorporation of PA nanofibers into dental composite materials on their mechanical properties is evaluated. In extant research, the focus is primarily given to the strategies for improving the quality of polymethyl methyl acrylate used in the production of dental prostheses by using electrospun fibers as reinforcement. The available findings indicate that the inclusion of silk fibroin nanofibers combined with poly (ethyleneimine) has the potential to increase the durability and comfort of PMMA-based dentures, thus improving patients’ quality of life [36].

### 3.1. FTIR Analysis

The goal of FTIR analyses was to assess and compare the chemical structure of two series of resin-based samples (as-prepared and saline-treated). As indicated by Figure 1, the FTIR spectra produced by the commercial dental composite material DiaFil 24h post-curing are identical to those obtained after 14 days of immersion in saline solution, with the exception of peak intensity. These findings indicate that PA nanofibers did not influence the curing process, while also affirming that the samples are stable and suitable for use under physiological conditions. As FTIR analyses of polymerized dental composite materials reinforced with nanofibers have not been previously conducted, this is a valuable observation, given that successful polymerization is the key determinant of clinical success.

In particular, suboptimal polymerization efficacy can result in altered biomechanical properties in terms of reduced material hardness, increased hydrolytic degradation, and diminished resistance to fracture and wear. As it can also lead to a significant release of residual monomer into the oral cavity, with adverse effects on the material biocompatibility, achieving sufficient bond strength between the material and the tooth structure is essential [37].

In the FTIR spectrum pertaining to the polymerized resin, the band related to the absorption of OH groups appears in the region spanning from 3200 to 3450 cm^−1^ and splits due to the intermolecular hydrogen bonds of OH groups. As the OH group absorption is accompanied by deformation vibrations (δOH), these are evident at 1460 cm^−1^. Asymmetric valence vibrations—ν_as_(CH_3_)—are also present in the spectrum, and are characterized by the bands at 2955 cm^−1^ and 2873 cm^−1^, while the band corresponding to C−H vibrations of the methylene group is located at 2929 cm^−1^. It is also evident that C=O valence vibrations from the ester COO−C group in TEGDMA are absorbed at 1721 cm^−1^ and, as expected, there is no significant shift in the position of this band due to saline treatment. The presence of aromatic structures in the as-prepared sample is confirmed by bands located at 1612 and 1511 cm^−1^, which are produced by vibrations of the C=C group of the aromatic ring. The C−O−C asymmetric valence vibrations of TEGDMA occur at 1297 cm^−1^, while the band located at 1168 cm^−1^ corresponds to the valence vibrations of the ester groups in the TEGDMA molecules comprising the dental resin. As a result of TEGDMA polymerization, the band related to the C−O−C symmetric valence vibrations shifts to 1103 cm^−1^ because polymerization introduces a larger number of C−O−C groups into the system. As DiaFil resin also includes UDMA, in the FTIR spectrum of the crosslinked resin, the band corresponding to the characteristic stretching of urethane bonds (N−H) at 3409 cm^−1^ is evident, as is that pertaining to the combination of urethane carbonyl groups (NH−CO−O) and ester carbonyl bonds (CO−O) at 1722 cm^−1^, along with the band related to C−N bonds, which extends to 1530 cm^−1^. The band located at 1249 cm^−1^ originates from the amide group III (COOC) vibrations. In the spectrum produced by the polymerized resin, a band at 1137 cm^−1^ corresponding to ν(C−O) vibrations emerges, which further amplifies the intensity of symmetric vibrations of the C−O−C group compared with the band at 945 cm^−1^ originating from the deformation vibrations of the C−H group in the aromatic ring. Given that the DiaFil resin-based composite material also contains Bis-GMA, this is reflected in the spectrum through the absorption of the ether C−O−C bond from bisphenol A at 1044 cm^−1^. The presence of an aromatic pair of substituted benzene rings from bisphenol A is confirmed by a band located at 829 cm^−1^, while bands at 1511 and 777 cm^−1^ represent vibrations related to =C−H groups from aromatic structures [37].

### 3.2. Morphology of Samples

To examine their morphology, after mechanical testing, the cross-section of all samples was examined under the scanning electron microscope. SEM analysis was chosen as it results in two-dimensional micrographs, while also benefitting from a wide range of magnifications, spanning from 10× to 100,000×. In particular, magnifications exceeding 1000× allow micromorphological characterization of the material structure. Although microcomputed tomography is a more accurate non-destructive three-dimensional quantification method, it is still highly expensive and requires much greater computer expertise compared to SEM analysis [38]. As shown in Figure 2, micrographs of pure resin were compared to those obtained for the reinforced composite resin materials 24 h after preparation. Figure 2a shows the fracture surface of pure resin, whereby the evident damage can be attributed to the brittle nature of its fracture. This fracture mode is expected because dental composite material is additionally reinforced with inorganic silicon dioxide filler, which further enhances the strength of pure resin. Figure 2b shows a sample with one layer of PA8 reinforcement, revealing a more uniform resin fracture morphology, which is indicative of greater toughness, and thus confirms the reinforcing effect of nanofibers. As no pronounced pouring of fibers from the resin structure is apparent, the resin and reinforcement appear to be highly compatible. Consequently, stress can be transferred from the resin to the PA layer, providing additional strength to the dental composite material, and supporting the findings yielded by the mechanical tests. The sample reinforced with the 16 μm-thick PA layer, denoted as res-PA16-res, is characterized by a more compact morphology of the fracture surface, which is expected due to the strengthening effect of nanofibers. As nanofibers are porous, during the crosslinking process, the non-crosslinked resin can penetrate through the reinforcement layer and form an interpenetrating network through the fibers. As a result, a much stronger bond between the resin and fiber layers is achieved, leading to a significant improvement in the mechanical properties of the dental material. Figure 2c further shows that no fibers are extracted from the resin matrix, and no large cracks on the fracture surface are evident, indicating that reinforcement with a 16 μm-thick PA layer is the most optimal. On the other hand, as evident from Figure 2d, the cross-section of the dental composite material with two layers of nanofiber reinforcement (res-PA8-res-PA8-res) exhibits larger cracks as well as areas with uniform fracture surfaces. Specifically, the region closer to the fibers is much more uniform, while damage is pronounced in the region farther from the reinforcement. This fracture pattern is attributed to the presence of three layers of dental composite material interspersed with two layers of fibers. Due to this arrangement, the inner regions benefit more from the reinforcement, giving rise to uneven stress distribution during deformation. Therefore, the mechanical properties of the res-PA8-res-PA8-res sample are comparable to those observed in the res-PA8-res sample.

The microcracks that could be observed on the micrographs shown in Figure 2 are not visible on the micrographs obtained after immersing the same samples in saline solution for 14 days at 37 °C. While improvements are evident in pure resin, as shown in Figure 3a, much greater enhancements are attained in samples reinforced with PA, as saline treatment rendered the samples more compact. As a result, in the res-PA8-res-PA8-res and res-PA16-ressystems, monomer penetration through fibers became more effective, leading to stronger crosslinks and the formation of interpenetrating networks, as indicated in Figure 3c,d. These findings are supported by the mechanical test results shown in Figure 4. Moreover, the PA nanofiber layer was unaffected by the physiological conditions simulated by sample immersion in a saline solution, which is expected, considering the properties of pure PA.

### 3.3. Mechanical Properties

The mechanical properties of the studied samples were examined by subjecting them to the three-point bending test, whereby the penetration (in mm) and flexural strength (in MPa) results are shown in Figure 4. It is evident that the reinforcement with PA nanofibers is beneficial, as penetration improved from 0.214 mm for the pure resin to 0.267 mm for resin with two 8 μm-thick PA nanofiber layers and further to 0.349 mm when a single 16 μm PA layer was employed. This result is expected given that in this configuration greater monomer penetration and stronger chemical bonds between resin layers can be achieved. Flexural strength is also improved with the PA addition, whereby more pronounced benefits are attained with two 8 μm-thick PA nanofiber layers and especially a single16 μm-thick PA reinforcement (which led to nearly 100% improvement in flexural strength compared to pure resin).

As expected, immersing samples for 14 days in saline solution at 37 °C was beneficial for both penetration and flexural strength, given that this treatment improved material density. Specifically, penetration increased from 0.008 mm for the pure resin to 0.03 mm for the res-PA16-res sample. Similarly, 3-fold and 4-fold improvements were observed in the flexural strength of res-PA8-res-PA8-res and res-PA16-res samples, respectively, compared to pure resin.

These findings are supported by the SEM results obtained in this work, as well as the findings reported by Nakano and colleagues, confirming the beneficial effect of adding nylon to dental composite materials in terms of increased flexural strength [39]. Similarly, Velo and colleagues reported that incorporating hybrid (inorganic−organic) nanofibers embedded with niobium pentoxide (Nb_2_O_5_) into self-adhesive resin cement has the capacity to provide reinforcement to dental materials. These authors further posited that niobium is particularly useful as a filler for improving the mechanical properties of dental composites [40].

### 3.4. Thermal Properties

The thermal properties of the studied samples were examined by producing DSC thermograms. It is evident from Figure 5 that, in the PA nanofibers, the glass transition occurs around 53 °C, and the cold crystallization peak is located at 93 °C, while the melting point is not shown, given that it occurs above 200 °C. However, the material behavior at such high temperatures was not of interest to the present investigation.

Figure 6 depicts the DSC thermograms pertaining to the dental composite resin-based samples obtained 24 h post-curing and after 14 days of immersion in saline solution. These results indicate that as-prepared as well as saline-treated reinforced samples have a lower glass transition temperature (Tg) compared to pure resin. Specifically, while pure resin has a Tg of 61.6 °C, each additional PA nanolayer decreases the Tg by about 2 °C, while further reduction is achieved in samples that have been immersed in saline solution for 14 days. It is also worth noting that some residual monomer is still present in the pure resin sample 24 h after curing, as evident from the thermogram at 160 °C, but is absent from samples that were stored in saline solution for 14 days. Moreover, as the mass of PA nanofibers present in the composite samples is too low, their thermal transitions are not visible in thermograms.

## 4. Conclusions

The work presented here demonstrates that the incorporation of electrospun polyamide fibers into resin-based dental materials improves their mechanical properties. In particular, the penetration and flexural strength can be modified by varying the thickness and number of reinforcing layers, which can be achieved by adjusting the process parameters and material design, respectively. Moreover, the results yielded by the FTIR spectroscopic analysis confirmed that the addition of PA fibers does not affect the crosslinking process within the dental composite material, which is very important for its clinical application. It is also noteworthy that, according to the SEM findings, the morphological properties of the dental material are nearly 100% improved through the addition of a thicker layer of PA reinforcement, as this enables the formation of a more compact structure, which results in improved mechanical properties of the material. On the other hand, for deriving the maximum benefit from the structure based on two PA reinforcement layers, the samples need to be treated with saline solution to increase material density and thus facilitate the formation of interpenetrating networks. Based on these findings, it can be summarized that the initial hypothesis guiding this research is supported, given that the PA-reinforced dental composite materials examined in this work are biologically acceptable and have the potential for use in clinical practice. Moreover, the work reported here has demonstrated that electrospinning is a facile method for producing different nanofibers that can be incorporated into resin-based dental composite materials to modify their mechanical properties.

## Figures and Tables

**Figure 1 polymers-15-02598-f001:**
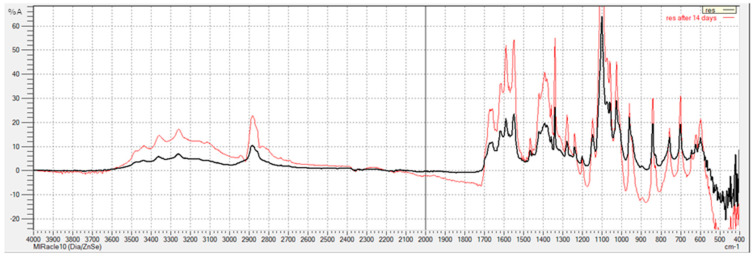
FTIR spectra of pure resin 24 h after curing (black line) and after 14 days of immersion in saline solution (red line).

**Figure 2 polymers-15-02598-f002:**
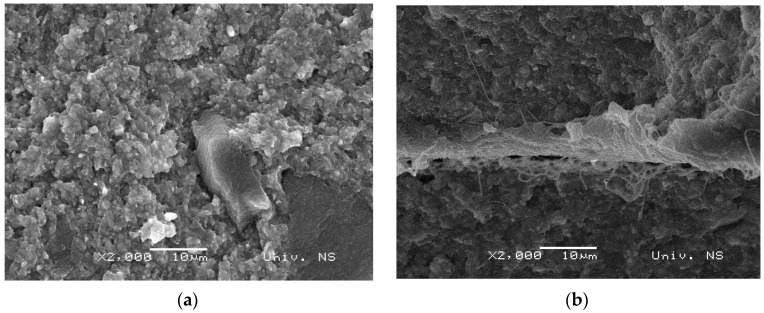
SEM micrographs of (**a**) pure resin, (**b**) res-PA8-res, (**c**) res-PA16-res, and (**d**) res-PA8-res-PA8-res samples (24 h after curing).

**Figure 3 polymers-15-02598-f003:**
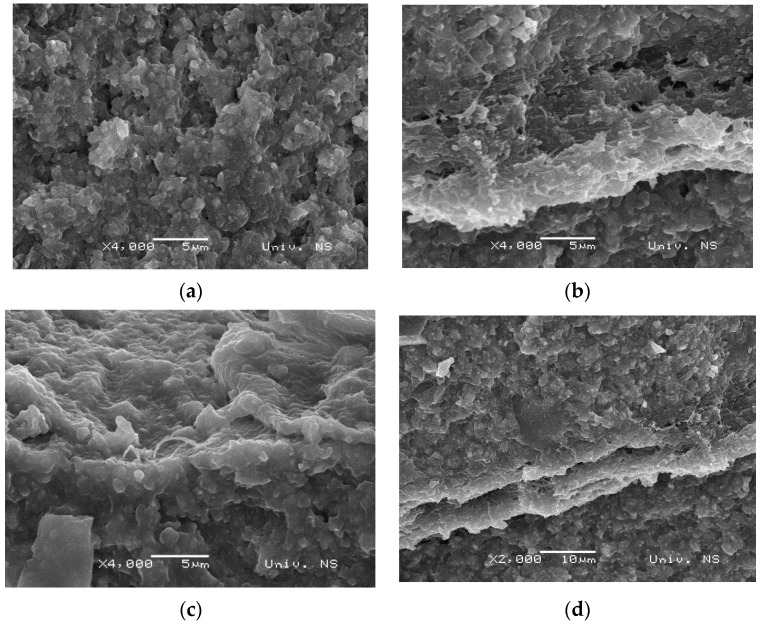
SEM micrographs of (**a**) pure resin, (**b**) res-PA8-res, (**c**) res-PA16-res and (**d**) res-PA8-res-PA8-res samples after 14 days of immersion in saline solution.

**Figure 4 polymers-15-02598-f004:**
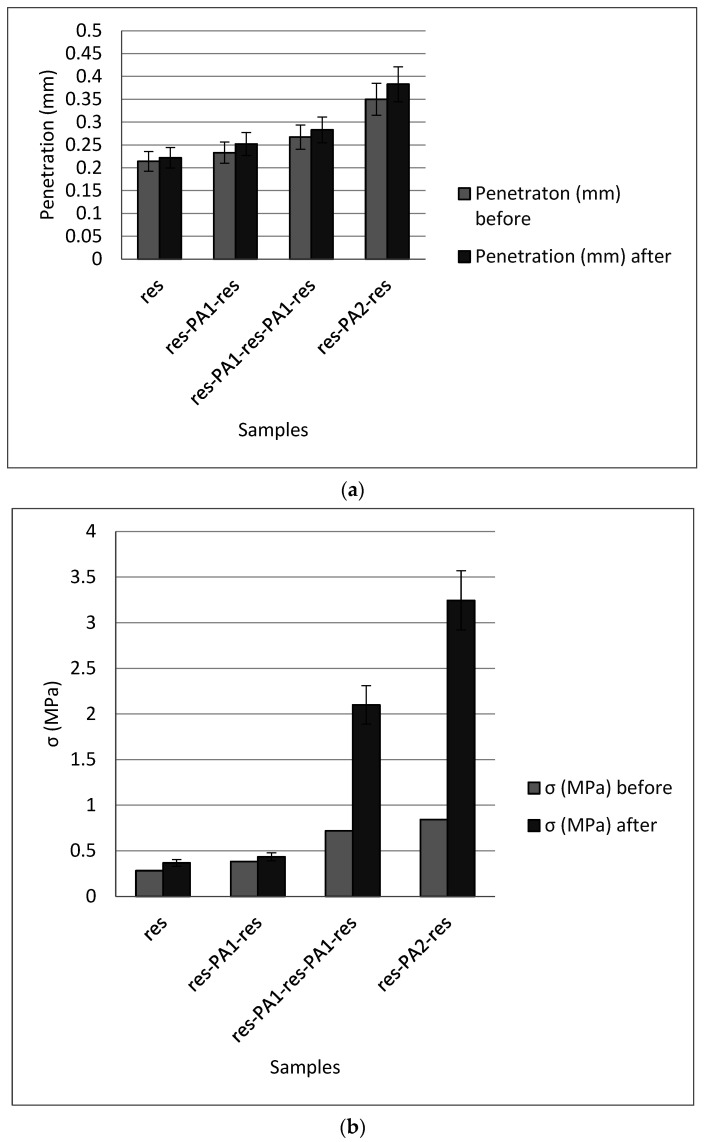
Mechanical properties of the pure resin, res-PA8-res, res-PA16-res and res-PA8-res-PA8-res samples in terms of (**a**) penetration depth and (**b**) flexural strength before and after 14 days of immersion in saline solution.

**Figure 5 polymers-15-02598-f005:**
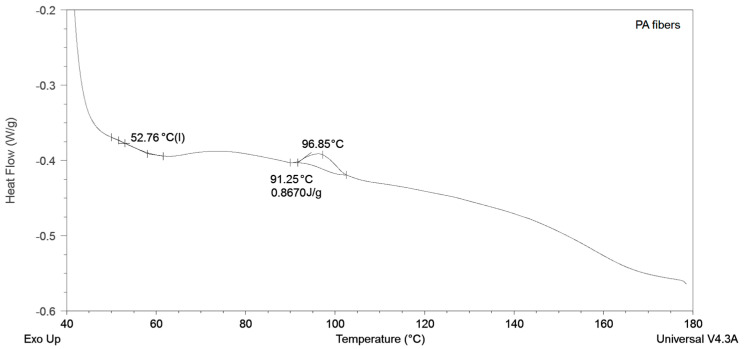
Thermal properties of PA nanofibers.

**Figure 6 polymers-15-02598-f006:**
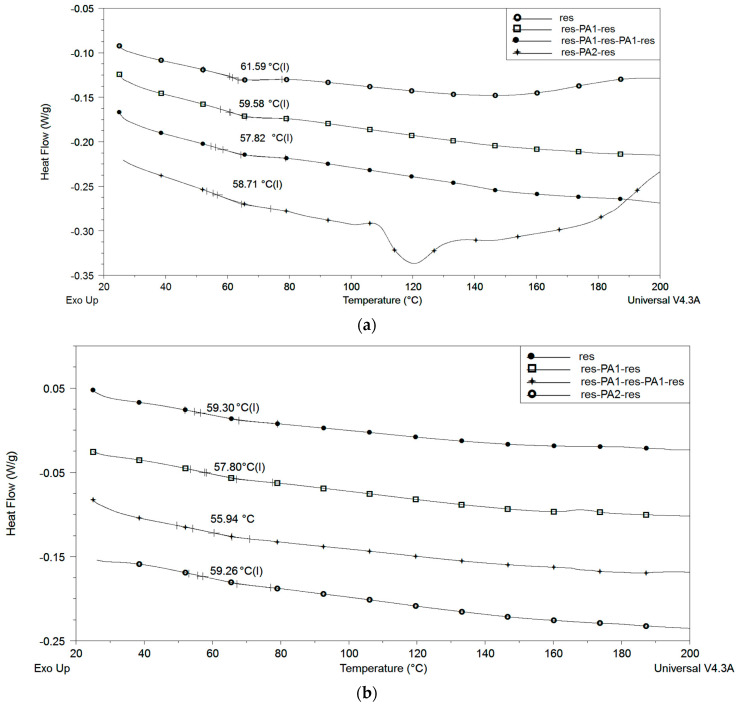
DSC thermograms of the pure resin, res-PA8-res, res-PA16-res and res-PA8-res-PA8-res samples (**a**) 24 h post-curing and (**b**) after 14 days of immersion in saline solution.

**Table 1 polymers-15-02598-t001:** Composition of DiaFil dental composite resin-based restorative material.

DiaFil Constituent	Synonym	Content (%)
Urethane dimethacrylate	UDMA	8−12
Bisphenol A glycidyl methacrylate	Bis-GMA	6−10
Triethylene glycol dimethacrylate	TEGDMA	1−4
Barium alumino silicate	Barium aluminosilicate	65−75
Silicon dioxide	Silicon dioxide	5−10
Camphorquinone	7,7-trimethyl-3-dione(+/−)-bicyclo[2.2.1]heptane-1(C amphorquinone)	0.01−0.02
Dibutylhydroxy toluene	2,6-di-butyl-4-Methylphenol(BHT)	0.001−0.002
Iron oxide	Iron Oxide	0.0001−0.01

**Table 2 polymers-15-02598-t002:** Samples used in the experiments.

Sample	Number of Resin Layers	Thickness of PA Layer	Number of PA Layers
res	1	-	-
res-PA8-res *	2	8 μm	1
res-PA16-res **	2	16 μm	1
res-PA8-res-PA8-res	3	8 μm	2

* res-PA8-res–sample with one PA nanofiber layer of 8 μm thickness placed between two resin layers. ** res-PA16-res –sample with one PA nanofiber layer of 16 μm thickness placed between two resin layers.

## Data Availability

Not applicable.

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
