# Peer review of "Development of Light-Polymerized Dental Composite Resin Reinforced with Electrospun Polyamide Layers"

_polymers, 2023, doi:10.3390/polym15122598_

Round 1

Reviewer 1 Report

Dear Auhtor ,

This is nice work showed use of polymer processing method of electrospinning.I have few comments and sugeestion

1.Manuscript lack constant and coherent flow ?

2.Title is characterization ?

3.Objective is reinforcement of dental composite ?

4.If possible remove characterization from title ?

5.What is novelty?

6.Conclusion need to rewrite?

7.Significance of study ?

8.Need to add dynamic mechanical analysis and TGA if possible?

Can be improved at certain place

Reviewer 2 Report

Dear Authors.

I found your work interesting. Manuscript: Characterization of light-polymerized dental composite resin reinforced with electrospun polyamide layers, raises an important issue, but the form of work needs improvement.

I have some revisions to propose to you to improve your work. Please refer to the following comments:

- Abstract: Please add 1 or 2 sentences of introduction so that the reader can get into the topic of the work.

- Abstract: Please add 1 or 2 sentences as conclusions to the work

- Introduction: lines 101-103: please add information about polyaramide fibers to reinforce the composite materials, You can use: Szalewski, L.; Kamińska, A.; Wallner, E.; Batkowska, J.; Warda, T.; Wójcik, D.; Borowicz, J. Degradation of a Micro-Hybrid Dental Composite Reinforced with Polyaramide Fiber under the Influence of Cyclic Loads. Appl. Sci. 2020, 10, 7296. https://doi.org/10.3390/app10207296

- Lines 121-132: Please rewrite this section as null hypotheses that you want to confirm/deny during the study

- line 153- how many samples were prepared?

- line 154- all samples had the same dimensions 2x2x2 mm? 

- line 156: What were the thicknesses of the layers of the composite resin?

- lines 157-158: what duration, mode and power of polyemrization was used? Was the distance from the lamp tip to the sample always the same for each sample preparation?

- line 164: how were the samples stored for 24 h?

- line 192 - what was the distance – L

- line 193- Were there any samples that differed in size from others and were therefore excluded at this stage?

- Conclusion- This is a repetition of the results, please rewrite this fragment so that these are the conclusions of the study and not the results themselves.

Reviewer 3 Report

The manuscript describes the reinforcement of dental composite resins with electrospun poly(amide), PA, nanofibers in order to obtain improved dental restoration materials.

The topic is of clinical relevance although the scope is incremental and “trial and error” of differing reinforcements.

There are mixed terminology: the initial introduction of restorative material later became dental cement.

The manuscript was not well prepared: although introduction also introduced reinforcement with nanoparticles and nanotubes, the study only focused on use of nanofibers.

Reproducibility is low due to the missing full product information of each instrument used. Please provide this in the format: (serial No. brand, manufacture, city, country).

The figures are not well prepared and appear insignificant. The axis labels and markings are not visible and the data lines are very thin. SEM micrographs do not have arrows indicating where the PA is.

There lacks statistical analysis to verify the “significant improvement”.

The discussion is not in-depth on the bonding and reinforcement mechanism of the PA.

English usage needs to be improved as there are various grammatical and spelling mistakes.  

Thus a major revision is recommended before further consideration.

Some more for consideration:

1. contraction should be replaced with polymerization shrinkage; longevity and lifespan mean the same so delete one.

2. electrospinned ->electrospun

3. L32, shown -> showed

4. L177-178, rewrite.

5. A recent study highlighting the interfacial bonding between organic matrix and inorganic filler particles in another dental composite material should be cited together with [3], doi: 10.1039/D1MA01002F

English usage needs to be improved as there are various grammatical and spelling mistakes. 

Round 2

Reviewer 2 Report

Dear Authors,

Thank you for responding to my comments.
I have no more comments for article.

Author Response

Thank you for such a comprehensive and detailed analysis of our manuscript and for the beneficial comments and recommendations, as addressing them has certainly improved the quality of our work. We appreciate this positive feedback.

Reviewer 3 Report

The authors have addressed the majority of the points rasied.

Strictly speaking, "significant" can not be used without statistical analyses. If the authors do not intend to do statistical analyses, replace significant with percent %, for example  in  L331, "...leading to a % improvement..." But this compromises the scientific soundness of the study. Wouldn't you want to confirm the addition of nanofibers has induced significant changes before carrying out future investigations?

Fig. 4, please add the error bars.

The statement "as the images presented in our paper were software-generated, their quality is beyond the influence of the manuscript authors" is not factual. Experience has taught us that the software also outputs raw data that we can plot using our preferred plotting program. Please see if you can do this, make the plots thicker, data points bigger and use larger font size for the axis labels and insets. 

SEM images should be aligned better.

DSC is missing some information: which gas was used at which flow rate, which reference was used as the baseline?

Correct typo "surmised".
